# Non-Coding RNAs in Myasthenia Gravis: From Immune Regulation to Personalized Medicine

**DOI:** 10.3390/cells13181550

**Published:** 2024-09-14

**Authors:** Nicola Iacomino, Maria Cristina Tarasco, Alessia Berni, Jacopo Ronchi, Renato Mantegazza, Paola Cavalcante, Maria Foti

**Affiliations:** 1Neurology 4–Neuroimmunology and Neuromuscolar Diseases, Fondazione IRCCS Istituto Neurologico Carlo Besta, 20133 Milan, Italy; nicola.iacomino@istituto-besta.it (N.I.); mariacristina.tarasco@istituto-besta.it (M.C.T.); alessia.berni@istituto-besta.it (A.B.); renato.mantegazza@istituto-besta.it (R.M.); 2Ph.D. Program in Neuroscience, University of Milano-Bicocca, 20900 Monza, Italy; jacopo.ronchi@unimib.it; 3Department of Medicine and Surgery, University of Milano-Bicocca, 20900 Monza, Italy; 4BicOMICs, University of Milano-Bicocca, 20900 Monza, Italy

**Keywords:** autoimmunity, long non-coding RNAs, microRNAs, myasthenia gravis

## Abstract

Myasthenia gravis (MG) is an antibody-mediated autoimmune disorder characterized by altered neuromuscular transmission, which causes weakness and fatigability in the skeletal muscles. The etiology of MG is complex, being associated with multiple genetic and environmental factors. Over recent years, progress has been made in understanding the immunological alterations implicated in the disease, but the exact pathogenesis still needs to be elucidated. A pathogenic interplay between innate immunity and autoimmunity contributes to the intra-thymic MG development. Epigenetic changes are critically involved in both innate and adaptive immune response regulation. They can act as (i) pathological factors besides genetic predisposition and (ii) co-factors contributing to disease phenotypes or patient-specific disease course/outcomes. This article reviews the role of non-coding RNAs (ncRNAs) as epigenetic factors implicated in MG. Particular attention is dedicated to microRNAs (miRNAs), whose expression is altered in MG patients’ thymuses and circulating blood. The long ncRNA (lncRNA) contribution to MG, although not fully characterized yet, is also discussed. By summarizing the most recent and fast-growing findings on ncRNAs in MG, we highlight the therapeutic potential of these molecules for achieving immune regulation and their value as biomarkers for the development of personalized medicine approaches to improve disease care.

## 1. Introduction

Myasthenia gravis (MG) is an autoimmune disease primarily caused by antibodies against the acetylcholine receptor (AChR) located on the postsynaptic membrane of the neuromuscular junction (NMJ). Less frequently, pathogenetic antibodies are directed to the muscle-specific tyrosine kinase (MuSK) or to the low-density lipoprotein receptor-related protein 4 (LRP4), two proteins involved in AChR clustering and muscle contraction [1]. A small proportion of patients have no specific autoantibodies to the known autoantigens (seronegative MG, or triple negative MG), thus requiring neurophysiological confirmation of diagnoses [1].

The autoimmune attack on the NMJ results in fluctuating weakness and early fatigability in skeletal muscles, which are the main disease symptoms [2].

MG has a complex or multifactorial etiology, resulting from interactions among multiple genetic, epigenetic, and environmental factors [2,3].

Epigenetic modifications play an important role in gene regulation, cell differentiation, tissue homeostasis, and susceptibility to diseases. They include DNA methylation, chromatin remodeling, non-coding RNAs (ncRNAs), and histone modifications [4]. NcRNAs, comprising microRNAs (miRNAs) and long non-coding RNAs (lncRNAs), modulate the expression of protein-coding genes involved in a variety of biological processes, including the immune response [5,6].

Over the last decade, the influence of ncRNAs on innate and adaptive immunity has been extensively investigated, especially in relationship with autoimmune disorders [7,8]. In MG, ncRNAs, particularly miRNAs, have increasingly emerged as key factors in the disease pathogenesis and potential biomarkers for patients’ stratification [9].

This article provides an overview of the current knowledge on the involvement of miRNAs and lncRNAs in MG. We discuss the role of these molecules as immunoregulatory factors and possible biomarkers associated with the different disease subgroups, and we highlight the new perspectives offered by the translation of ncRNA research into the MG clinical practice.

## 2. Myasthenia Gravis: Clinical Features and Immunopathology

MG is a heterogeneous disease, the clinical variability of which allows for patients’ stratification into distinct disease subgroups according to the following features: (i) autoantibodies profile (AChR-, MuSK-, LRP4-, or triple seronegative MG); (ii) symptoms’ distribution (ocular versus generalized MG); (iii) age at disease onset (early-onset, EOMG, versus late-onset, LOMG: <50 versus >50 years); and (iv) presence or absence of thymoma [2,10].

About 80–90% of diagnosed patients carry anti-AChR antibodies in the serum that are clearly responsible for the observed disease symptoms. Pathogenicity of anti-AChR antibodies is mainly due to complement system activation at the NMJ; additional pathogenic mechanisms are AChR blockage or antigenic modulation, the last occurring when an antibody cross-links with two antigen molecules, triggering cellular signals that cause the antigen endocytosis and degradation [11]. A variable proportion of patients have pathogenic autoantibodies against two different NMJ components, including MuSK (5–8% of patients) and LRP4 (7–33% of patients; 5–20% of anti-MuSK antibody-positive and 7.5% of anti-AChR antibody-positive patients), two proteins involved in AChR clustering [1,2]. A small proportion of patients have no specific autoantibodies detectable by classical diagnostic tests for AChR and MuSK antibodies (i.e., double seronegative MG patients) [1]. In recent years, the use of cell-based assays has made it possible to detect anti-AChR or -MuSK autoantibodies, or both, in patients previously defined as double seronegative by classical methods, also allowing for the discovery of anti-LRP4 antibodies [1].

Antibodies against striational antigens, i.e., titin and ryanodine receptors, are frequently found in patients with thymoma and, to a lesser extent, in LOMG [1,2]. They are of uncertain pathogenicity and are useful as biomarkers; indeed, their presence is predictive of thymoma in EOMG clinical phenotypes [12]. Patients without specific autoantibodies detectable by the classical diagnostic tests are referred to as triple seronegative [1,2].

The main clinical hallmarks of MG are fluctuating weakness and fatigability in the ocular, bulbar, and skeletal muscles [2,10]. In most patients, the first symptoms to appear involve ocular muscles, which then progress to generalized MG (gMG) within 2–5 years [13].

The thymus is widely recognized as the main site of the autoimmunity initiation reaction in AChR-MG. Most AChR-MG patients present thymic pathological alterations, including follicular hyperplasia (~60%), characterized by the presence of ectopic germinal centers (GCs) forming follicles in the thymic medulla, and thymoma (up to 30%), a typical thymic epithelial tumor [3]. Chronic inflammation, uncontrolled activation of innate immune responses and type I interferon (IFN) production, likely triggered by pathogen infections, are immunological alterations contributing to the intra-thymic pathogenesis of AChR-MG associated with follicular hyperplasia [3,14]. B cell dysfunction, with abnormal circulating B cell activation and lymphoid neogenesis into the thymus, are pathogenetic AChR-MG hallmarks that are relevant to the development of a sustained autoimmune response [15].

Immunoregulatory defects related with T cells have also been well documented in MG patients; they are mainly caused by impairment of both regulatory T cells (Tregs) and conventional T cells and by an altered balance between Tregs and pathogenic T helper 17 (Th17) cells observed in follicular hyperplastic thymuses and the peripheral blood of MG patients [16]. A low frequency of Tregs has also been reported in patients with MG carrying thymomas [16,17]. Abnormal T cell selection and failure in Treg differentiation are intra-thymoma alterations implicated in the development of thymoma-associated MG, along with an autoimmune regulator (AIRE) deficiency that is suspected to cause the dangerous presentation of muscle autoantigens locally expressed into the thymus, and consequent autoreactivity [17,18,19].

## 3. Non-Coding RNAs as Regulators of Immune Responses

NcRNAs, such as miRNAs, lncRNAs and circulating miRNAs, have emerged as important regulators of both innate and adaptive response, as they are involved in the control of immune system cell biology and function. Since their first description in 1993 [20,21], studies on ncRNAs in immunity have greatly increased [5,6,7,22]. Dysregulated ncRNAs can influence multiple immune-related biological processes, such as immune homeostasis, immune tolerance, immune cell development, differentiation and proliferation, and the imbalance between pro-inflammatory/anti-inflammatory cytokines [22,23,24]. For example, many ncRNAs have been shown to be involved in NF-κB, JAK-STAT, and MAPK signaling, thus influencing a wide range of inflammation-related pathways and participating in a large spectrum of inflammatory and autoimmune diseases [8,25].

MiRNAs are small single-stranded RNAs of approximately 22 nucleotides that are able to post-transcriptionally regulate target mRNAs and cellular functions. They are also detectable in the extracellular space and can be found in any human body fluids, where they are called circulating miRNAs [26]. They are secreted through membrane-enclosed extracellular vesicles, such as microvesicles and exosomes, which serve as carriers for cellular communication both in the microenvironment and in distant tissues. By this mechanism, miRNAs may act as epigenetic regulators of the gene expression in distal cells [27,28].

Due to their pivotal role in fine-tuning the functionality of the immune system, several miRNAs have been defined as “immuno-miRs” [29]. The fundamental impact of immuno-miRs on both innate and adaptive immune systems includes the modulation of key inflammatory signaling pathways (e.g., NF-kB) and their downstream effectors, as well as the regulation of proliferation, differentiation, and functions of hematopoietic cells [5,29,30,31].

MiR-146a-5p, miR-150-5p, and miR-155-5p are among the most studied immuno-miRs, and their contribution in the context of the immune system has been well characterized.

As a well-known key modulator of both the innate and adaptive immune system, miR-146a-5p can be considered as a molecular bridge between innate and adaptive immunity. Specifically, the miRNA acts as a potent inhibitor of Toll-like receptor (TLR) pathways via the repression of key transducers of the NF-kB signaling (i.e., IRAK1, TRAF6), thus dampening the immune response in a negative feedback mechanism to avoid persistent innate immune activation and inflammation [32,33]. In parallel, miR-146a-5p modulates T and B cell function; in particular, it promotes Treg-mediated immune tolerance by targeting STAT1, limits accumulation of the T follicular helper (Tfh) cells by targeting ICOS, and prevents B cell proliferation, differentiation, and GC formation by targeting c-REL and Fas [34,35,36,37].

MiR-150-5p is an immuno-miR selectively expressed in hematopoietic stem cells in a developmental stage-dependent pattern, as it is absent during early proliferative stages whereas its expression increases during T and B cell differentiation [38]. The miRNA modulates the expression of several target genes, such as genes encoding the Notch3 receptor family member [39] and the transcription factors c-Myb [40] and STAT1 [41], all being involved in multiple steps of lymphocyte differentiation and immune system cell activation.

First described in B cell lymphoma, miR-155-5p is an immuno-miR abundantly expressed in lymphoid organs, such as the thymus and spleen, and in hematopoietic stem cells [42,43]. MiR-155-5p mediates inflammatory responses and promotes the development of Th17 and Th1 cell subsets in inflammatory contexts [44]. It participates in the GC reaction by contributing to an optimal T cell-dependent antibody response [45]. Moreover, it is involved in innate immunity, as its expression is enhanced upon immune cell stimulation via TLR signaling and, in turn, regulates the activation of macrophages and the production of inflammatory cytokines [46,47].

Due to their critical immunoregulatory functions, miRNAs are clearly involved in autoimmune diseases [8]. Dysregulated expression of miR-146a-5p [48,49,50], miR-150-5p [51], and miR-155-5p [52] has been reported in different autoimmune conditions, including systemic lupus erythematosus (SLE), rheumatoid arthritis (RA), multiple sclerosis (MS), and Sjögren’s syndrome.

In SLE, aberrant expression of other miRNAs, such as miR-31, miR-145 and miR-142-3p, has been associated with the pathogenesis [53]. The involvement of miR-326, miR-128-3p, miR-27a, miR-320b, miR-132-3p, miR-181c-5p, miR-34a, miR-142-3p, and additional miRNAs has been described in MS [54]. Furthermore, several miRNAs, mainly miR-21, miR-140, miR-223, miR-29, miR-34 and miR-125, were shown to be dysregulated in RA [55].

LncRNAs are non-coding transcripts longer than 200 nucleotides that are located in the nucleus or cytoplasm, where they act as critical transcriptional and post-transcriptional regulators of biological processes [56].

In the last decade, the role of lncRNAs in the landscape of the immune system has been explored, and accumulating evidence has shown lncRNAs’ ability to modulate immune responses. Indeed, lncRNAs are involved in innate immunity regulation and inflammatory, antiviral processes, as well as in T and B cell differentiation, activation, and responses through different molecular mechanisms [57].

Among the immune-related lncRNAs, MALAT-1, one of the most well-studied lncRNAs, has been shown to influence a wide spectrum of cellular processes, including cell multiplication and specialization, as well as apoptosis and inflammation [58]. In macrophages, MALAT-1 acts as an NF-kB pathway modulator [59], whereas other observations showed its contribution in the phenotypical shaping of CD4+ T cells and dendritic cells [60,61]. Recent studies suggested that MALAT-1, as with other lncRNAs, exerts its immunoregulatory functions by interacting with miRNAs (e.g., miR-30b, miR-15/16) and modulating their functionality [62,63,64] within lncRNA/mRNA/miRNA networks.

Since its first description as an essential factor for T cell growth control [65], GAS5 has emerged as an important lncRNA regulator in the proliferation, differentiation, and activation of innate and adaptive immune system cells, particularly macrophages and Th17 cells [66,67]. Additional lncRNAs involved in immunity are (i) Morrbid, which controls the survival of neutrophils, eosinophils, and monocytes in response to pro-survival cytokines by repressing the Bcl2l11 transcription [68]; (ii) IFNG-AS1, also known as NeST, which positively regulates the expression of IFN-γ in T cells and natural killer cells [69,70]; (iii) GATA3, another regulator of T cell immunological function and differentiation [71]; NRON, a repressor of the nuclear factors of activated T cells (NFATs) that are transcription factors regulating the expression of genes involved in T cell development, activation, and differentiation [72,73].

As miRNAs, lncRNAs have been recognized as playing a role in the development of autoimmune diseases by several mechanisms, including the modulation of T cell differentiation and Th17/Treg ratios [74]. The interaction between Th17 and Treg cells was found to be regulated by the lncRNA MEG3 in Behçet’s disease and SLE [75]. In SLE, IL21-AS1 was found to be involved in interleukin-2 and Tfh regulatory cell activation [76]. Furthermore, dysregulated expression of MALAT-1 and ANRIL has been associated with the pathogenesis of SLE and correlated with disease activity, thereby indicating these two lncRNAs as potential biomarkers for SLE [53].

MALAT-1 has been associated with the etiology of several autoimmune conditions, including SLE, RA and MS, via epigenetic modifications, alternative splicing, modulation of gene expression networks and interactions with miRNAs [58,77]. In patients affected by these diseases, MALAT-1 exhibits markedly elevated levels, thus impacting on immune system cell function and differentiation processes, as well as cytokine profiles and activities [58,77].

In conclusion, with the role of ncRNAs in the immune system becoming clearer, this knowledge will translate into the understanding of how these types of molecules affect the initiation, progression, and resolution of autoimmune diseases. Due to their immunoregulatory effects, targeting ncRNAs, including both miRNAs and lncRNAs, is an attractive and promising molecular approach to achieving the control of autoimmunity and counteracting pathogenic immune responses in the context of these diseases. Treatments based on miRNA-targeting drugs have made significant progress in preclinical and clinical testing for some diseases [78]. This field is particularly challenging due to the need to rigorously understand the function of the miRNA candidates in relationship with the different cell types in which they are expressed and to develop specific and efficient strategies for miRNA delivery in the target cells. If these difficulties are overcome, RNA therapeutics that are able to modulate immune response could revolutionize the therapeutic scenario of autoimmune diseases.

## 4. MiRNAs in MG

The role of miRNAs in MG has received increasing consideration in the last two decades, as the disease pathogenesis has not yet been fully elucidated into molecular details and the study of miRNAs has the potential to aid in identifying pathogenic events and pathways underlying loss of tolerance and induction of the autoimmune response. Moreover, understanding the functional role of miRNAs in MG may provide new avenues for patient’s stratification in disease subgroups and the development of tailored miRNA-based therapeutic approaches. A number of studies revealed dysregulated miRNA expression in the thymus and peripheral blood of MG patients, as well as the impact of miRNA dysregulation on a variety of immune-related pathways (Table 1, Figure 1), thus indicating the role of miRNAs as pathogenic factors and potential targets for innovative treatments.

### 4.1. MiRNAs in MG Thymuses: Role in Intra-Thymic Disease Pathogenesis

#### 4.1.1. Hyperplastic MG Thymuses

Functional and morphological thymic abnormalities are frequently observed in AChR-MG patients, including thymic hyperplasia and thymoma [3,17]. Thymic hyperplasia is mainly characterized by the presence of B cell infiltrates in the thymic medulla, herein leading to the formation of ectopic GCs, supported by active neo-angiogenic processes [15,105]. An MG thymus contains all the cellular and molecular components required to develop the anti-AChR autoimmune response, including the autoantigen, which is expressed in thymic epithelial cells (TECs) and muscle-like myoid cells, autoreactive T and B cells, and plasma cells able to produce autoantibodies [3]. The role of the thymus in the pathogenesis of the disease is supported by evidence that thymectomy, which is mandatory in thymomatous patients, is able to improve MG symptoms along with prednisone compared to prednisone alone in patients with thymic hyperplasia [106]. Loss of immunoregulation with defective Tregs, chronic inflammation and persistent TLR-mediated innate immune responses to viral infections have been demonstrated in the hyperplastic thymus of AChR-MG patients and are thought to be responsible for autosensitization and autoimmunity [3,14,15,16]. However, the exact intra-thymic molecular alterations underlying dysregulated innate and adaptive immune responses, as well as uncontrolled autoreactivity, are still under investigation.

The first global miRNome analysis of an MG thymus was carried out in EOMG patients with different degrees of thymic hyperplasia by miRNA arrays, revealing 61 differentially expressed (DE) mature miRNAs between MG and control thymuses, including 24 up- and 37 down-regulated miRNAs [85]. Among them, miR-125a-5p and miR-7-5p were validated as miRNAs up- and down-regulated in MG thymuses by qPCR. Of interest, the increased levels of miR-125a-5p inversely correlated with those of its target gene, WD repeat-containing protein 1 (WDR1), a protein involved in auto-inflammatory processes [107]. Moreover, an inverse correlation between the levels of miR-7-5p and those of its target gene CCL21, encoding a B cell-attracting chemokine implicated in GC formation, was observed [85]. This correlation was confirmed by functional in vitro experiments in TECs [85], suggesting that the already known CCL21 overexpression in MG thymuses [105] may be a consequence of miR-7-5p down-regulation, and hence that miR-7-5p dysregulation may contribute to GC development and hyperplastic changes in the thymus of MG patients.

Of interest, a study from Sengupta and colleagues [108] investigated the miRNAs’ role in GC formation and maintenance, revealing a different transcriptional profile between GC-positive and GC-negative thymic samples from MG patients. Specifically, miRNA expression profiling performed by microarrays showed an alteration of several miRNAs involved in inflammation and immune system pathways, such as cytokine regulation, humoral and cellular immunity, apoptosis, and cell proliferation, between the two groups of thymuses.MiR-139-5p and miR-452-5p levels were found to be down-regulated in GC-positive thymuses, where the regulator of the G-protein signaling 13 (RGS13) gene was overexpressed [108]. This miRNA/target gene interaction was validated by in vitro experiments. Since RGS13 is a G-coupled protein that regulates the chemokine responsiveness of GC B cells, these results suggested the involvement of the identified miRNAs in GC formation in MG thymuses [108].

Another study performed in hyperplastic MG thymuses reported the dysregulation of 33 miRNAs in these tissues compared to normal thymuses by an miRNA microarray chip [87]. Among the DE miRNAs, miR-548k was expressed at a significantly lower level in MG than control thymuses, whereas its target gene CXCL13, encoding a key chemokine in intra-thymic MG pathogenesis [109], mediating B cell homing and motility in secondary lymphoid tissues, was strongly overexpressed. As demonstrated in this study, miR-548k functionally regulates CXCL13, and thus its down-regulation in MG thymuses may play a role in the disease pathogenesis by enhancing B cell recruitment and GC formation [87].

Several studies have also focused on disclosing the role of the major immuno-miRs in intra-thymic MG pathogenesis. MiR-150-5p expression levels were found to be increased in the thymus of EOMG MG patients compared to controls, and this overexpression was particularly evident in patients with a high degree of thymic hyperplasia [79]. MiR-150-5p levels positively correlate with those of CD19 mRNA in MG thymuses, and the miRNA was localized in the B cells of the mantle zone of GCs by in situ hybridization. Mechanistically, in vitro experiments showed that miR-150-5p modulates the expression of the proto-oncogene MYB [79], a key transcription factor involved in hematopoiesis, thus suggesting an miR-150-5p contribution to T and B cell survival and GC formation.

Recently, a down-regulation of miR-146a in the thymus of EOMG patients has been described, and this down-regulation was inversely correlated with increased expression of the miRNA target genes IRAK1, c-REL, and ICOS [88]. Laser-capture microdissection showed that miR-146a deficiency was restricted in the GC-surrounding medulla and, accordingly, IRAK1 was overexpressed in the thymic stroma. Double fluorescence analysis revealed an increased amount of IRAK1-expressing myeloid dendritic cells and macrophages around GCs, whereas c-REL and ICOS overexpression was associated with GCs. Due to the key role of IRAK1 in TLR-mediated innate immune responses, and of c-REL and ICOS in GC development, these results suggested that defective miR-146a expression could contribute to persistent TLR activation, lack of inflammation resolution, and hyperplastic changes in MG thymuses [88]. Interestingly, treatment with corticosteroids was found to increase miR-146a levels in peripheral blood cells (PBMCs) in vitro, thus suggesting that these drugs may induce control of autoimmunity in MG via miR-146a [88].

#### 4.1.2. MG Thymomas

Thymomas are uncommon and malignant neoplasms derived from TECs associated with many paraneoplastic disorders and autoimmune diseases, mainly MG, which develops in approximately 30–45% of thymoma patients [110,111]. The classification of World Health Organization (WHO) recognizes type A, AB, B1, B2, and B3 thymomas based on the lymphocyte content and the epithelial cell features [112]. Type B2 thymoma is the histological type most frequently associated with MG, followed by types AB and B1 [113].

As mentioned above, AIRE deficiency, reduction in myoid cells, and failure in Treg generation are the main intra-thymoma alterations predisposing to the development of MG [17].

In the last decade, growing evidence revealed a possible contribution of miRNAs to thymoma-associated MG.

A genome-wide study performed by microarrays showed a differential expression of about 137 miRNAs in normal tissue adjacent to thymoma in MG patients compared to those with a normal thymus [86]. As observed in hyperplastic MG thymuses [85], miR-125a-5p was found to be markedly overexpressed in MG samples, and its abnormal expression was functionally linked to the down-regulation of FOXP3, the master transcription factor implicated in Treg generation and functionality, suggesting a contribution of this miRNA to Treg deficiency in MG thymomas. MiR-376a and miR-376c were found to be down-regulated in MG tissues [89]. Since these miRNAs were found to be repressed in AIRE-silenced TECs [89], the data indicated a relationship between defective AIRE expression, down-regulation of miR-376a and miR-376c, and susceptibility to autoimmunity development in thymoma [89].

Another study exploring miRNA expression in MG thymomas showed elevation of miR-19b-5p in the neoplastic thymic tissue of MG patients compared to normal thymuses, as well as the ability of this miRNA to post-transcriptionally inhibit the expression of thymic stromal lymphopoietin (TSLP), which indeed was down-regulated in MG thymomas [90]. Since TSLP has a prominent role in the regulation of thymic cell lineages, these data suggested a contribution of miR-19b-5p to T cell dysregulation and Treg deficiency in the thymomas of MG patients [90].

A key miRNA implicated in MG associated with thymomas is miR-20b. Xin and colleagues [91] showed that expression levels of this miRNA were significantly decreased in MG thymoma tissues compared to adjacent non-tumor tissues, resulting in an increased proliferation and activation of T cells through the NFAT5/CAMTA1 dependent pathway. Both NFAT5 and CAMTA1 are miR-20b target genes; the first is vital for cell cycle progression and proliferation of T cells, and the second is an integrator and effector of calcium signaling modulating the NFAT5 activity [91]. Since miR-20b was found to suppress T cell proliferation and activation [91], as well as Th17 differentiation, by targeting RORγ and STAT3 [114], its low levels could well underlie the abnormal balance between pathogenic Th17 and protective Tregs in MG thymomas.

### 4.2. Dysregulated miRNAs in MG Peripheral Blood

MiRNA profiles have been extensively studied in the PBMCs and serum of MG patients, revealing a number of altered miRNAs that could represent non-invasive disease biomarkers (Figure 1).

The first microarray-based study of miRNA expression in MG and control PBMCs revealed 44 DE miRNAs, highlighting down-regulation of all the members of the let-7 family in MG compared to control cells [94]. Let-7c expression was further investigated, demonstrating that it might participate in MG pathogenesis by directly modulating IL-10 expression, a key orchestrator of the immune system [94]. A higher number of AChR-reactive IL-10 mRNA-expressing PBMCs was previously found in more MG patients than in controls [115], and IL-10-secreting cells tended to be higher in patients with generalized MG than in patients with the ocular disease [116], thus suggesting a link between let-7c down-regulation and IL-10 production in MG.

In a subsequent study, miRNA profiling by microarrays, followed by miRNA validation via qPCR, in PBMCs revealed a significant down-regulation of miR-320a in MG patients compared to controls [95]. MAPK1, which is involved in inflammatory response and immune defense, was found to be a direct target of miR-320a. Moreover, the miRNA down-regulation induced the overexpression of pro-inflammatory cytokines by promoting COX-2 expression in a process modulated by ERK/NF-κB pathways, thus indicating a possible role of miR-320a in regulating inflammatory cytokine production in MG [95].

Another miRNA found to be down-regulated in MG PBMCs is miR-145. The expression of this miRNA was reduced in PBMCs and CD4+CD25- (effector) T cells from both experimental autoimmune MG (EAMG) rats and MG patients [96]. Of interest, CD28 and NFATc1, two crucial molecules involved in T cell activation and expansion, were target genes of miR-145, and the miRNA overexpression in AChR-specific CD4+ T cells suppressed NFATc1 and IL-17 expression, thus suggesting miR-145 deficiency as a mechanism contributing to abnormal T cell activation and pathogenic Th17 response in MG [96]. In these studies, no information on the autoantibody status of patients was provided in order to associate the identified miRNAs with the MG serological subgroups.

Altered circulating miRNAs have also been described in pediatric MG, a disease subgroup affecting children. In the recent study of Zhu and colleagues [117], small extracellular vesicle (sEV)-miRNAs were isolated from the plasma of pediatric MG patients (aged 7.42 ± 5 years), including patients negative and positive for anti-AChR antibodies, and controls (age 5.12 ± 2.44 years) and were subjected to RNA sequencing. Several DE miRNAs were found, including 50 miRNAs DE between MG patients and controls, 48 DE between AChR-MG patients and controls, and 39 DE between anti-AChR antibody-negative MG patients and controls. Twenty-four altered miRNAs were commonly DE in the three comparison sets, including the above-mentioned miR-125a-5p, already associated with adult MG, and miR-143-3p, which was validated by qPCR as being down-regulated in pediatric MG patients compared to controls [117]. Although the literature data on ncRNAs in pediatric and juvenile MG are still limited, this study points out the involvement of miRNAs in pediatric MG and, specifically, the role of sEV-miRNAs as circulating biomarkers and possible future therapeutic targets in this disease subgroup [117].

#### 4.2.1. AChR-MG-Associated miRNAs

AChR-MG patients represent the most frequent and studied MG clinical subgroup, and this subgroup can be further subdivided according to additional clinical features, such as age at disease onset (EOMG vs. LOMG) and presence or absence of thymoma [1,2].

A whole transcriptomic analysis performed in PBMCs from AChR-EOMG patients and healthy controls showed the differential expression of 11 miRNA precursors and their predicted target genes, which were distinctive of an inflammatory signature [97]. The identified miRNAs, namely miR-612, miR-3654, and miR-3651, were validated as mature miRNAs, and the analysis confirmed their significant up-regulation in the PBMCs of patients compared to controls. Correlation analyses of miRNA/target gene pairs revealed their potential relationship, thus suggesting a contribution of the dysregulated miRNAs in the AChR-EOMG inflammatory autoimmune process [97].

The miR-181 family has been reported to be involved in innate and adaptive immune response by regulating the myeloid and lymphoid lineage differentiation, particularly T cell differentiation [118,119]. Decreased miR-181c expression in PBMCs was found to correlate with elevated serum levels of IL-7 and IL-17 in AChR-MG patients, thus suggesting a contribution of this miRNA to loss of immune regulation and Th17 cell responses via production of pathogenic pro-inflammatory cytokines in MG [98].

The expression of the miR-15 cluster was also decreased in AChR-MG PBMCs in association with increased expression of CXCL10, also named interferon (IFN)-γ-induced protein 10 (IP-10); this protein acts as chemoattractant for activated T cells and is a functional direct target of miR-15a [99]. The miR-15a decrease was postulated to abnormally activate immune response, while its increase could reduce CXCL10 expression and alleviate abnormal T cell activation in MG [99].

The immuno-miRs miR-150-5p [79] and miR-146a-5p [88] were found to be, respectively, down- and up-regulated in AChR-MG compared to healthy control PBMCs. MiR-150-5p was mainly down-regulated in CD4+ T cells compared to healthy control cells, suggesting an miRNA release in the serum from these cells when activated [79]. Indeed, miR-150-5p levels are increased in AChR-MG patients’ serum [80]. Contrariwise, miR-146a-5p was shown to be increased in AChR-MG PBMCs, in line with its reduction in the serum of the same patients compared to healthy controls [88].

Dysregulated miRNAs have been reported to normalize in the PBMCs of AChR-MG patients undergoing immunosuppressive therapy. For instance, corticosteroid treatment restores the miR-15a levels in steroid-responsive MG patients [99]. In addition, the miR-146a-5p level was found to be normalized in corticosteroid-treated AChR-MG PBMCs and sera [88], supporting the hypothesis of a relationship between this miRNA and the therapeutic effects of corticosteroids.

Extracellular circulating RNAs have been widely studied in MG serum. An early example of miRNA profiling performed with serum showed that both miR-150-5p and miR-21-5p expression levels were elevated, whereas miR-27a-3p and miR-30e-5p levels were reduced in AChR-positive EOMG patients compared to healthy subjects [80]. Of interest, stable immunosuppression therapy (≥6 months) was able to normalize (i.e., reduce) the amount of both miR-150-5p and miR-21-5p in patients’ serums (41% and 25% lower than the baseline for miR-150-5p and miR-21-5p, respectively) [81]. The reduced serum levels of these two miRNAs were observed in AChR-MG patients included in the MGTX trial and underwent thymectomy along with prednisone treatment at 24 months post-surgery compared with baseline and with patients treated only with prednisone [82].

The involvement of serum miR-150-5p up-regulation in MG pathogenesis was further explored, and possible molecular mechanisms have been postulated. Increased serum levels of miR-150-5p were found to positively correlate with those of IL-10 in patients with generalized MG, with a reduction in both miR-150-5p and IL-10 being seen after treatment [120]. These data supported the role of the miRNA in the cytokine network, and particularly in the regulation of IL-10, a key anti-inflammatory cytokine with multifaceted functions, including antibody production [120]. In addition, serum miR-150-5p was described to act as a modulator of peripheral blood cell behavior, particularly cell proliferation, by targeting MYB, as reported above [79].

MiRNA expression was analyzed in different AChR-MG subgroups in order to evaluate the association of specific miRNAs with the disease phenotype. Serum miR-15b, miR-122, miR-140-3p, miR-185, miR-192, and miR-20b were down-regulated in both EOMG and LOMG patients compared to controls, suggesting common altered mechanisms [92]. MiR-885-5p was another miRNA that was down-regulated in LOMG patients, whereas, in MG thymoma patients, only miR-15b was significantly down-regulated compared to controls, likely because of the small number of patients in this group [92]. Down-regulation of miR-20b in serum was confirmed in another study and correlated negatively with Quantitative Myasthenia Gravis (QMG) in pre-treated patients; its levels were normalized after treatment with prednisone, although they continued to be negatively correlated with QMG [93]. Functional studies revealed IL-8 and IL-25 as miR-20b target genes, and indeed their expression was decreased following treatment with prednisone [93].

An extended investigation performed longitudinally in follow-up samples from a large cohort of LOMG patients, irrespective of autoantibody subtype, revealed increased levels of miR-150-5p, miR-21-5p, and miR-30e-5p in LOMG compared to control sera [83]. Of note, the levels of these miRNAs decreased in parallel with clinical improvement after initiation of immunosuppression and were positively correlated with the clinical MG composite score (MGC), suggesting a value as disease biomarkers for LOMG, as previously reported for EOMG [83]. Serum levels of miR-150-5p and miR-21-5p were higher in general if compared with ocular LOMG [83], according to their correlation with MG severity.

In regards to the ocular and generalized disease forms, there are no known biomarkers that can be used to predict the conversion from the first to the second form, although AChR-MG patients are likely to be at higher risk of conversion than AChR antibody-seronegative patients [121]. Two miRNAs were found to be significantly higher in patients (i.e., the majority LOMG) who developed generalized MG compared to ocular LOMG patients, namely miR-30e-5p and miR-150-5p. Of these miRNAs, miR-30e-5p had 96% sensitivity for differentiating ocular MG and gMG in all ocular patients and 100% in LOMG ocular patients, indicating its potential predictive value, at disease presentation, as a biomarker of the risk of generalization for patients with ocular MG [84].

The overall findings indicated a contribution of miR-150-5p, miR-21-5p, and miR-30e-5p to AChR-MG, as well as their possible role as disease biomarkers. Interestingly, to test the biomarker value of these three miRNAs, their stability was assessed in a recent study in which the short-term changes in the miRNA serum levels were analyzed weekly in MG patients with unchanged medications undergoing follow-up visits for 1 month [122]. Changes in patient-reported outcome measures, including the MG activities of daily living (MG-ADL), MG quality-of-life-15 (MG-QoL15), and Fatigue Severity Scale (FSS), were also assessed [122]. Intra-patient levels of miR-30e-5p and miR-150-5p were found to be stable, as were the MG-QoL15 and FSS scores, whereas miR-21-5p showed a significant reduction from week 1 to 3 [122]. Demonstration of miR-30e-5p and miR-150-5p short-term stability supported the role of these two miRNAs as biomarkers for MG, stimulating prospective multi-center studies with longer follow-up periods to validate this role [122].

#### 4.2.2. MuSK-MG-Associated miRNAs

MuSK-MG differs from AChR-MG for both the pathogenic effect of MuSK autoantibodies and the more severe disease manifestations, with worse clinical outcomes [1,2]. Accordingly, these differences between the two subgroups are well recapitulated by different miRNA expression profiles.

MuSK-MG-associated miRNAs have been poorly studied in PBMCs. Using next generation sequencing, 96 down- and 5 up-regulated miRNAs were identified in PBMCs from MuSK-MG patients compared to healthy controls [101]. The top six significant miRNAs from the discovery set were validated using RT-qPCR, revealing four of them, i.e., miR-340-5p, miR-160b-5p, miR-27a-3p, and miR-15a-3p, as the most DE miRNAs in MuSK-MG patients compared to controls, although their levels did not correlate with anti-MuSK antibody titer or QMG scores [101].

Interestingly, MuSK-MG, but not AChR-MG [101], has been strictly associated with the let-7 miRNA family [101]. Specifically, miRNA analysis using a PCR Panel covering 179 miRNAs, followed by individual miRNA validation, revealed altered expression of two of these miRNAs, let-7a-5p and let-7f-5p, in the serum of MuSK-MG patients compared to controls, although their levels were not correlated with disease severity or antibody titers [101]. In a separate study, miRNA microarrays disclosed 21 miRNAs which were significantly overexpressed, and 23 which were underexpressed, in the PBMCs of MG patients compared with healthy controls [94], with miRNAs belonging to the let-7 family (except mir-98 and let-7e) being significantly underexpressed, particularly let-7c, which was the best representative decreased miRNA. The autoantibody profiles of patients were not reported in this study to understand whether let-7 family miRNAs are also altered in the PBMCs of MuSK-MG patients or of both MuSK-MG and AChR-MG patients [94]. IL-10 was found to be a target of let-7c, and its expression negatively correlated with let-7c levels in PBMCs, thus revealing the contribution of the miRNA to MG via IL-10 regulation [94]. Let-7 miRNA family members have been implicated in immune response by regulating T and B cell function [123] and, particularly, by inhibiting differentiation of Th17 cells [102], thus implying its defective expression in PBMCs as a pathogenic alteration in the autoimmune disease context. For instance, let-7f-5p inhibits Th17 differentiation through targeting STAT3 and has been found to be down-regulated in the CD4+ T cells of patients with MS [102].

Additional miRNAs DE in MuSK-MG patients were miR-151a-3p and miR-423-5p [101], both implicated in cancers and, particularly, in cell proliferation [124,125]. These miRNAs were overexpressed in the serum of patients versus controls [101], pointing out a possible link, to be explored, between their increase and abnormal proliferation of immune system cells that may favor autoimmune response initiation or perpetuation. Of note, miR-151a-3p was reduced in MuSK-MG patients treated with rituximab, a monoclonal antibody targeting CD20 expressed in B cells, after a six-month treatment in which clinical severity scores were reduced [103], leading to the hypothesizing of a relationship between MuSK-MG-associated miR-151a-3p increase, B cells, and disease pathogenesis.

A PCR miRNA Panel was used to assess circulating miRNAs in the plasma of MuSK-MG patients and healthy controls, revealing three DE miRNAs, including miR-210-3p and miR-324-3p, which were strongly decreased, and miR-328-3p, which was elevated, in patients versus controls [104]. Data for miR-210-3p and miR-324-3p were further validated, suggesting a contribution of these miRNAs in MuSK-MG [104]. Both miR-210-3p and miR-324-3p have been described as having tumor suppressive functions [126,127], further suggesting a relationship between MuSK-MG-associated miRNAs and possible increased immune system cell proliferation.

## 5. Role of lncRNAs in MG

Studies on the involvement of ncRNAs in MG are mostly focused on miRNAs, while the contribution of lncRNAs as molecular factors implicated in the disease has not been thoroughly investigated yet, although their profile has been demonstrated as being altered in the thymus and peripheral blood of MG patients (Table 2, Figure 1).

LncRNAs regulate different points of miRNA biogenesis and act as competing endogenous RNAs for miRNAs [128]; therefore, dysregulated miRNA expression observed in both the thymus and peripheral blood of MG patients could be related to lncRNA changes.

**Table 2 cells-13-01550-t002:** Summary of the literature data on lncRNAs associated with MG.

LncRNA	Autoantibody Status	Alteration: Up- (↑) or Down-Regulated (↓)	Target mRNA/Interacting miRNA	References
XLOC_003810	AChR+	↑ In thymic CD4+ T cells of MG thymoma patients ↑ In thymus of both MG patients with and without thymoma ↑ In PBMCs from MG thymoma patients	-	[129,130]
XLOC_006297, XLOC_007052, XLOC_002588	AChR+	↑ In MG compared to non-MG thymomas	PRSS56, PPP3R1, DUSP26 and PLCG1	[131]
LINC00452	AChR+	↑ In MG compared to non-MG thymomas	↓ miR-204 ↑ CHST4	[132]
oebiotech_11933	AChR+	↑ In PBMCs of MG thymoma patients	-	[133]
XLOC 003810, XLOC 005780, ENSG00000259354.1	AChR+	↑ In PBMCs of non-thymoma MG patients	-	[134]
XLOC 000734, ATP6VOE2-AS1, ENSG00000250850.2	AChR+	↓ In PBMCs of non-thymoma MG patients	-	[134]
IFNG-AS1	AChR+ and AChR-	↓ In PBMCs of MG patients (including patients with normal, hyperplastic and thymoma thymus)	CD40L, Tbet	[135]
MALAT-1	n.a.	↓ In PBMCs of EO and LOMG patients	miR-338-3p MSL2	[136]
GAS5	AChR+	↓ In PBMCs of non-thymoma EOMG patients ↓ In CD4+ T cells from MG patients	miR-23a IL-10	[137,138]
SNHG16	n.a.	↑ In PBMCs of MG patients	Let-7c-5p IL-10	[139]
HCG18	n.a.	↑ In PBMCs of MG patients	miR-145-5p CD28	[140]
NR_104677.1, ENST00000583253.1, NR_046098.1, NR_022008.1, ENST00000581362.1	AChR+	↑ In serum exosomes of MG patients	Network with 14 MG-associated miRNAs and 30 mRNAs	[141]

AChR: acetylcholine receptor; CD40L: CD40 Ligand CHST4; Carbohydrate Sulfotransferase 4; DUSP26: Dual Specificity Phosphatase 26; IL-: interleukin; MSL2: Male-Specific Lethal-2 Homolog; n.a.: not available; PBMCs: peripheral blood mononuclear cells; PLCG1: Phospholipase C Gamma 1; PRSS56: Serine Protease 56; PPP3R1: Protein Phosphatase 3 Regulatory Subunit B, Alpha; Tbet: T-Box Transcription Factor 21.

### 5.1. Dysregulated lncRNAs in MG Thymuses

The role of lncRNAs in MG thymuses has only been addressed by a few studies, mainly performed on the thymomas. Aberrant expression of lncRNAs could underlie the imbalance of Th17/Treg cells, which are critically involved in the intra-thymic pathogenesis of MG, particularly in the disease subgroup associated with thymoma. A study of Niu and colleagues [129] showed overexpression of the lncRNA XLOC_003810 in thymic CD4+ T cells from MG thymomas compared to controls cells and provided a demonstration that the lncRNA overexpression enhanced the transdifferentiation of Tregs toward Th17 cells, whereas its silencing attenuated the Th17/Tregs imbalance. The XLOC_003810 contribution to T cell dysregulation was confirmed in a separate study of the same group in which XLOC_003810 up-regulation was shown in MG thymuses both with and without thymoma, along with enhanced levels of inflammatory Th17 cytokines [130]. Collectively, these data indicated a role of XLOC_003810 in regulating T cell activation and Th17/Treg ratios in MG thymoma.

To investigate lncRNA expression in thymoma patients with and without MG, Ke and colleagues assessed the transcriptome profiles in these patients by microarrays and analyzed the co-expression relationship of aberrant lncRNAs and mRNAs [131]. A huge number of genes, both lncRNAs (e.g., XLOC_006297, XLOC_007052, and XLOC_002588) and mRNAs (e.g., PRSS56, PPP3R1, DUSP26, and PLCG1), were dysregulated between MG and non-MG thymomas, and subsequent gene ontology (GO) analysis and function annotation revealed their involvement in different biological processes, especially dephosphorylation and hydrolysis, which are essential for thymocyte survival during intra-thymic selection [142]. Thus, lncRNAs were further implied in abnormal T cell selection, which is known to be critically involved in autoimmunity development in MG thymomas [17]. In this context, a recent study of Wang and colleagues [132] revealed a cross-talk between the lncRNA LINC00452 and miR-204 in regulating thymic Tregs in MG thymoma. They performed RNA sequencing and constructed a competitive endogenous RNA (ceRNA) network with DE RNAs between MG and non-MG thymomas, followed by data validation via qPCR. The overall results showed that LINC00452, miR-204-3p, miR-204-5p, and CHST4, a gene expressed in high endothelial venules and crucial for lymphocyte homing, were DE between the two groups of patients and that these RNAs constitute a molecular axis likely implicated in Treg impairment in thymoma associated with MG [132].

### 5.2. Dysregulated lncRNAs in MG Peripheral Blood

The first study providing evidence of the involvement of lncRNAs in MG analyzed lncRNA and mRNA expression in the PBMCs of MG patients with and without thymoma and healthy controls by high throughput microarrays [133]. The results indicated that lncRNA and mRNA profiles were significantly different between MG patients with and without thymoma; moreover, several lncRNAs and mRNAs were commonly altered in MG patients with and without thymoma versus healthy controls. The lncRNA oebiotech_11933 exhibited the highest up-regulation among MG patients with thymoma versus healthy controls, and GO analysis showed an association of this lncRNA with lymphocyte immune cell proliferation and cancer development pathways [133]. LncRNA and mRNA co-expression analysis was performed, and, for co-expressed and aberrant lncRNA genes, GO and pathway analysis revealed the involvement of genes altered in thymoma MG patients versus healthy controls in several functions that are important for MG pathogenesis, such as ‘cellular response to IFN-γ’, ‘chemokine receptor binding’, and ‘lymphocyte proliferation’ [133]. In MG patients without thymoma versus healthy controls, ‘chemokine receptor binding’, ‘cytokine-cytokine receptor interaction’, and ‘platelet alpha granule’ were the most enriched GO terms. In the thymoma versus non-thymoma MG group, the most enriched GO terms were ‘chemokine receptor binding’ and ‘cytokine-cytokine receptor interaction’. A number of the disturbed lncRNAs were validated, leading to the suggestion that MG-associated lncRNAs could be implicated in the regulation of lymphocyte differentiation and proliferation and cytokine/chemokine networks [133].

A further study identified 1561 up-regulated and 1034 down-regulated lncRNAs, along with 921 up-regulated mRNAs and 806 down-regulated mRNAs, in the PBMCs of non-thymoma MG patients compared to controls by microarrays [134]. Several GO terms, including ‘nucleic acid transcription factor activity’, ‘inflammatory response’, ‘regulation of leukocyte activation’, ‘lymphocyte proliferation’, and ‘regulation of B cell proliferation’, were enriched in gene lists. Among DE lncRNAs, 33 were predicted to have 31 cis-regulated genes and 65 were predicted to have 45 trans-regulated target genes, according to a co-expression network analysis [134]. Although only six lncRNAs were validated (i.e., XLOC 003810, XLOC 005780, ENSG00000259354.1, XLOC 000734, ATP6VOE2-AS1, ENSG00000250850.2), the results suggested a potential correlation among the identified DE lncRNAs and MG [134].

Several studies addressed more in detail the potential role of individual immune-related lncRNAs in MG pathogenesis. IFNG-AS1 was found to be down-regulated in the PBMCs of MG patients compared to controls, and its levels were negatively correlated with the QMG score and serum anti-AChR antibody titer [135]. Functional in vitro studies demonstrated the lncRNA ability to reduce the expression levels of CD40L and the transcription factor Tbet in CD4+ MG T cells and to decrease the Th1/Treg cell subset ratio, thus suggesting that IFNG-AS1 down-regulation may be involved in CD4+ T cell-mediated immune responses in MG [135].

MALAT-1 is another lncRNA implicated in MG, since it was found to be down-regulated in PBMCs from MG patients compared to controls [136]. This lncRNA is involved in cell proliferation; thus, its down-regulation might indicate increased immune system cell proliferation in MG patients. Moreover, by acting as a ceRNA for miR-338-3p, MALAT-1 has been found to directly induce the expression of male-specific lethal 2 homolog (MSL2), which is involved in chromatin organization, maintenance of normal histone profiles and DNA damage response [136], thus indicating a contribution of the MALAT-1/MSL2/miR-338-3p molecular axis in MG.

Similarly, GAS5 has been shown to be down-regulated in the PBMCs of MG patients, and data have suggested that its overexpression directly up-regulates IL-10 expression to improve the disease, thus suggesting a potential role of GAS5 as a potential therapeutic target or biomarker of treatment outcome in MG [137].

Some lncRNAs and RNA transcripts can have ceRNA activities by actively competing for miRNA binding through sets of conserved miRNA response sequences, thus forming a large-scale regulatory network across the transcriptome [143]. Although lncRNAs have been recognized as ceRNAs in MG, such as MALAT-1 through the MALAT-1/miR-338-3p/MSL2 axis [136], very few studies have focused on the interactions among the lncRNAs, miRNAs, and mRNAs underlying the immune dysregulation of MG. The first ceRNA network constructed in MG identified the lncRNA SNHG16 as a regulatory factor implicated in the disease [139]. SNHG16, overexpressed in MG PBMCs, was found to be a target gene of let-7c-5p, able to regulate the expression of IL-10 by sponging let-7c-5p in a ceRNA manner; moreover, this lncRNA was shown to inhibit cell apoptosis and promote cell proliferation by sponging let-7c-5p [139]. Lately, by using specific integrative data analysis tools, Li and colleagues constructed a lncRNA-mediated module-associated ceRNA network to systematically explore the regulatory roles of lncRNAs in MG. They pointed out the HCG18/miR-145-5p/CD28 ceRNA axis, as implicated in the disease [140].

Specifically, HCG18 was overexpressed in MG patients and was found to serve as ceRNA to regulate CD28 expression by competitively binding miR-145-5p. Since CD28 is essential for T cell activation, these data suggested a contribution of HCG18 to MG via modulation of T cell functions [140].

Similarly, a recent study of the same group showed that OIP5-AS1 may exert the same effect on T cells through the modulation of IL-7 expression by sponging miR-181c-5p [144]. Finally, Xu and colleagues confirmed GAS5 down-regulation in MG CD4+ T cells, where this lncRNA was described to inhibit differentiation of Th17 cells through negative regulation of miR-23a, thus eventually contributing to the imbalance of the Th17/Tregs ratio in MG [138].

LncRNAs have also been studied in serum. High throughput RNA sequencing of exosomal lncRNAs was performed in the serum of MG patients and controls, and the best five lncRNAs (NR_104677.1, ENST00000583253.1, NR_046098.1, NR_022008.1, and ENST00000581362.1) were validated as being significantly increased in MG serum exosomes compared to controls [141]. Among them, NR_046098.1 levels correlated with MG severity. Enrichment analysis showed that DE genes annotated for these lncRNAs were within the functional classes of immune-related pathways, including the T cell receptor signaling pathway [141]. A specific lncRNA/miRNA/mRNA regulatory network, including the 5 lncRNAs, 14 MG-related miRNAs, and 30 mRNAs, was constructed, but the main functional role of these lncRNAs was not addressed to link the network with specific MG-associated immunological mechanisms [141].

Single nucleotide polymorphisms (SNPs) in lncRNAs may disturb miRNA binding sites. Wang and colleagues [145] proposed a catalog of risk genes and miRNAs for MG and constructed a lncRNA-mediated ceRNA network for the disease based on multi-step computational strategies. They identified risk pathways for MG and lncRNA-SNPs that could affect lncRNA/miRNA/mRNA interactions and regulate these risk pathways. A potentially significant role for the MAPK signaling pathway (hsa04010) was identified, and six high-risk genes (BCL2, KRAS, MAPK14, VEGFA, RAF1, and ESR1) and associated lncRNA-SNPs in MG patients were revealed. By mining lncRNA-SNPs in high-risk genes for MG, the authors proposed the four most significant potential mechanisms for lncRNA-SNP-gene-pathway effects: rs138852863/EPB41L4A-AS1 → VEGFA → hsa04010 (MAPK signaling pathway); rs2516515/HCP5 → RAF1 → hsa05205 (proteoglycans in cancer); rs17177030/MCM3AP-AS1 → BCL2 → hsa01521 (EGFR tyrosine kinase inhibitor resistance)/hsa01522 (endocrine resistance); rs2476391/DLEU2 → ESR1 → hsa05205 (proteoglycans in cancer) [145]. The MAPK signaling pathway was identified as a key risk pathway for MG, and, within this pathway, VEGFA was an important regulatory factor for cell survival [145]. Nevertheless, further studies are needed to translate these findings into biological functions by testing patients’ samples and/or preclinical models, as well as to verify that these networks are indeed dysregulated in MG.

## 6. NcRNAs as Biomarkers for Personalised Medicine

Autoimmune diseases, such as MG, are complex and clinically heterogeneous conditions whose effective management depends on many factors, such as early diagnosis, personalized group risk, disease phenotype stratification, and the possibility of monitoring responses to therapy. The identification of robust biomarkers to predict and measure therapeutic responses, prevent disease progression and exacerbation, and assess the development of drug resistance in a reproducible and non-invasive way, is a critical medical need. The adoption of biomarker-guided personalized medicine (PM) approaches able to tailor preventive measures and medical treatments to the characteristics of each patient, including the genetic and epigenetics profiles, promises to significantly increase the therapeutic success and cost-effectiveness ratio of therapies in the context of autoimmune diseases, including MG [146]. Prognostic, diagnostic and predictive biomarkers can indeed guide clinical decisions to ensure that the optimal therapeutic program is offered to each individual patient. Because they have a key regulatory role in immune system cell development and function, ncRNAs, particularly miRNAs, are ideal candidate biomarkers for PM introduction in clinical practice. Moreover, they have many features of optimal epigenetic biomarkers, as they are stable in body fluids; their expression profile is cell- or tissue-specific, or specific of precise biological stages; they can be easily assessed and are non-invasive if analyzed in circulating blood; their expression reflects specific pathophysiological conditions or diseases; their changes in circulating fluids or cell populations can be associated with clinical improvement upon treatments and can reflect individual responsiveness to drugs at early stages, thus allowing for therapeutic follow-ups [147].

MiRNAs have been demonstrated to be useful in stratifying patients based on MG subgroups (e.g., AChR-MG, MuSK-MG, EOMG, LOMG), as widely discussed above; moreover, the expression levels of some of them were found to change during treatment in relationship with symptom improvements [146]. Nevertheless, research regarding ncRNAs as PM biomarkers is still in its early stages in terms of MG. Interestingly, a deeper miRNome sequencing was performed in whole peripheral blood in a study cohort of Italian and Israeli AChR-MG patients, stratified as responders or non-responders to conventional immunosuppressive therapy [148]. MiRNA sequencing identified 41 DE miRNAs, and a cluster of three miRNAs—miR-323b-3p, miR-409-3p, and miR-485-3p (chr 14q32.31)—was validated as being down-regulated in non-responders compared to responder MG patients. Contrariwise, miR-181d-5p and -340-3p showed an opposite trend. GO enrichment analysis, along with mRNA target prediction and in silico modeling for the function of the identified miRNAs, disclosed the functional involvement of the five miRNAs in both immune response (i.e., neurotrophin TRK and Fc-epsilon receptor signaling pathways) and drug metabolism regulation [148]. ROC curve analysis showed sensitivity and specificity performance results indicative of miR-323b-3p, miR-409-3p, and miR-485-3p predictive values for responsiveness to immunosuppressive drugs in MG [148], thus suggesting their usefulness as a non-invasive molecular tool to monitor the efficacy of these drugs. Early miRNA-guided prediction of unresponsiveness, or of disease exacerbation upon treatment, could direct patients towards more effective therapies, such as those based on the new biological drugs.

## 7. Conclusions and Future Perspectives

The exact molecular events underlying autoimmunity initiation and chronicity in MG are not fully elucidated. MiRNAs and lncRNAs are critical gene expression regulators that are able to interact with each other and with mRNAs, resulting in the modulation of a plethora of pathophysiological processes, including innate and adaptive immune response. Their expression profiles are altered in the thymus, peripheral blood cells, and serum from MG patients (Table 1 and Table 2, Figure 1), in relationship with disease phenotype, clinical symptoms and response to therapies, thus highlighting the roles of ncRNAs as pathogenetic factors as well as biomarkers for the disease. The contribution of specific ncRNAs to the immunological alterations leading and sustaining MG supports the idea that RNA therapies targeting these molecules could provide new advanced molecular strategies to modulate the immune system and counteract autoimmunity. Due to their potent regulatory function, and their cell-, tissue- and disease-specific expressions, ncRNAs have tremendous therapeutic potential and represent ideal druggable targets to obtain normalization of MG-related genes to inhibit the autoimmune process. RNA therapeutics administered with appropriate vectors are emerging as efficient treatment strategies; several RNA drugs have been approved, and some of them are in phase III trials to treat rare and common diseases [149]. However, their potential is still poorly explored in the field of autoimmunity, pointing out the importance of further investigations to identify specific immunoregulatory lncRNA/miRNA/mRNA networks and reveal the in vitro and in vivo effects of their manipulation in inhibiting autoimmunity. A new scenery for MG treatment could be opened by the study of ncRNAs, since RNA-based treatments could be a valuable strategy to treat the disease, particularly in patients with refractory disease, intolerant to conventional immunosuppressive therapies, and non-responders to the new biological drugs. MG-associated immuno-miRs (e.g., miR-150-5p, -146a-5p, -21-5p) represent the most promising candidate ncRNAs to be tested as novel therapeutic targets for their immunoregulatory properties and potential to affect T and B cell function.

Biomarkers which are useful in developing PM strategies include genetic factors associated with distinct MG subgroups, or responses to the different drugs, and circulating proteins, whose profiles could enable monitoring of the disease course and drug effects in individual patients. This research area is still open. The potential use of ncRNAs as biomarkers able to guide therapeutic decisions also requires further studies. Although promising, the ncRNA-guided tailored approaches are still far from being adopted for MG, as well as for other autoimmune diseases, thus stimulating research towards the identification of reliable, stably expressed molecules reflecting patient-specific disease statuses and responses to the treatments to predict and monitor drug efficacy at individual levels within more effective PM treatment programs. Finally, the different MG-associated miRNAs, miR-150-5p, -30e-5p, and -21-5p, are suggested for further evaluation for their potential role as PM biomarkers in both AChR- and MuSK-MG patients, along with miR-20b for AChR-MG and let-7a-5p and -7f-5p for MuSK-MG patients. A positive correlation between these ncRNAs and the individual response to treatments via longitudinal analyses in wide patients’ cohorts is expected to provide useful data for future PM implementation in MG clinical practice.

## Figures and Tables

**Figure 1 cells-13-01550-f001:**
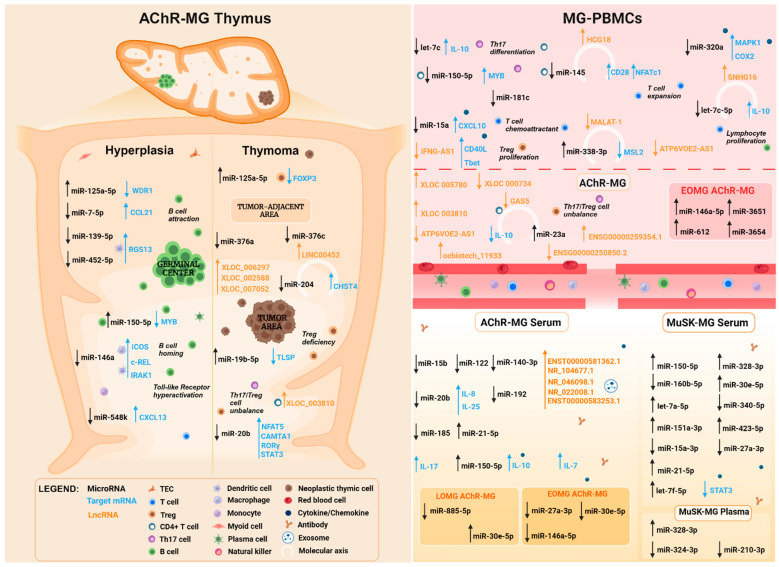
MicroRNAs, long non-coding RNAs, and related mRNAs known to be dysregulated in MG. A number of microRNAs (miRNAs, black), long non-coding RNAs (lncRNAs, orange), and related mRNAs (sky blue) were found to be dysregulated in the thymus, peripheral blood mononuclear cells (PBMCs), and serum/plasma of MG patients. Up- and down-regulated miRNAs are reported with arrows pointing up or down, along with implications of their dysregulation for T and B cell function and immune response (italics writing). MiRNAs, LncRNAs, and mRNAs interacting with each other and forming a molecular axis/triplet are represented with the semicircle. The left section illustrates ncRNAs, and some related mRNAs, altered in the thymus of AChR-MG patients, including hyperplastic MG thymuses, characterized by a germinal center presence in the thymic medulla, and MG thymoma, of which miRNAs are dysregulated in the tumoral area or in the normal tumor-adjacent area, are shown. The right upper and lower sections show molecules altered in MG PBMCs and serum/plasma, respectively. Boxes within these sections enclose miRNAs specifically altered in distinct disease subgroups, including early- (<50 years) and late- (>50 years) onset MG. NcRNAs altered in serum are subdivided according to the autoantibody status of MG patients (AChR- and MuSK-MG patients).

**Table 1 cells-13-01550-t001:** Summary of literature data on main miRNAs associated with MG.

MiRNA	Autoantibody Status	Alteration: Up- (↑) or Down-Regulated (↓)	Changes Related to Treatments	Target/Putative Target mRNAs	References
miR-150-5p	AChR+	↑ In thymus of EOMG patients with hyperplasia ↑ In serum of EOMG patients ↑ In serum of gMG compared to oMG patients ↓ In total PBMCs and CD4+ cells of EOMG patients ↑ In serum of LOMG	↓ In serum of patients after thymectomy and treatment with IS drugs in accordance with clinical improvement	MYB IL-10	[79,80,81,82,83,84]
miR-21-5p	AChR+	↑ In serum of EOMG and LOMG patients ↑ In serum of gMG compared to oMG patients	↓ In serum of patients after thymectomy and treatment with IS drugs in accordance with clinical improvement	-	[80,81,82,83]
miR-30e-5p	AChR+	↓ In serum of EOMG patients ↑ In serum of LOMG patients ↑ In serum of gMG compared to oMG patients	↓ In patients after treatment with IS drugs in accordance with to clinical improvement	-	[80,83,84]
miR-125a-5p	AChR+	↑ In thymus of EOMG patients with hyperplasia ↑ In normal tissue adjacent to MG thymoma	-	WDR1 FOXP3	[85,86]
miR-7-5p	AChR+	↓ In thymus of EOMG patients with hyperplasia	-	CCL21	[85]
miR-548k	AChR+	↓ In thymus of EOMG patients with hyperplasia	-	CXCL13	[87]
miR-146a-5p	AChR+	↓ In thymus and serum of EOMG patients ↑ In PBMCs of EOMG patients	↑ In thymus and serum of patients treated with IS drugs	IRAK1, c-REL, ICOS	[85,88]
miR-376a, miR-376c	AChR+	↓ In normal tissue adjacent to MG thymomas	-	Relationship with AIRE	[89]
miR-19b-5p	AChR+	↑ In MG thymoma	-	TSLP	[90]
miR-20b	AChR+	↓ In MG thymoma ↓ In serum of EOMG and LOMG patients	↓ In serum of patients treated with IS drugs	NFAT5, CAMTA1, RORγ, STAT3, IL-8, IL-25	[91,92,93]
let-7c	n.a.	↓ In PBMCs of MG patients	-	IL-10	[94]
miR-320a	n.a.	↓ In PBMCs of MG patients	-	MAPK1, COX-2	[95]
miR-145	n.a.	↓ In PBMCs and CD4+CD25- (effector) T cells of MG patients	-	NFATc1, CD28	[96]
miR-612, miR-3654 and miR-3651	AChR+	↑ In PBMCs of EOMG patients	-	-	[97]
miR-181c	AChR+	↓ In PBMCs of MG patients	-	-	[98]
miR-15a	AChR+	↓ In PBMCs of MG patients	↑ In patients treated with IS drugs	CXCL10	[99]
miR-15b, miR-140-3p, miR-185, miR-192	AChR+	↓ In serum of EOMG and LOMG patients	-	-	[92]
miR-27a-3p	AChR+	↓ In serum of EOMG AChR+ patients	-	-	[79,100]
miR-27a-3p, miR-340-5p, miR-160b-5p, miR-15a-3p	MuSK+	↓ In serum of MuSK+ patients	-	-	[100]
let-7a-5p	MuSK+	↑ In serum of patients	-	-	[101]
let-7f-5p	MuSK+	↑ In serum of patients	-	STAT3	[101,102]
miR-151a-3p	MuSK+	↑ In serum of MG patients	↓ In patients treated with RTX according with clinical improvement		[101,103]
miR-423-5p	MuSK+	↑ In serum of MG patients	-	-	[101]
miR-210-3p	MuSK+	↓ In plasma of MG patients	-	-	[104]
miR-324-3p	MuSK+	↓ In plasma of MG patients	-	-	[104]

AChR: acetylcholine receptor; AIRE: autoimmune regulator; CAMTA1: Calmodulin Binding Transcription Activator 1; CCL21: C-C Motif Chemokine Ligand 21; COX-2: Prostaglandin-Endoperoxide Synthase 2; c-REL: REL Proto-Oncogene, NF-KB Subunit; CXCL10: C-X-C Motif Chemokine Ligand 10; CXCL13: C-X-C Motif Chemokine Ligand 13; EOMG: early-onset (<50 years) MG; FOXP3: Forkhead Box P3; ICOS: Inducible T Cell Costimulator; IL-: interleukin-; IRAK1: Interleukin 1 Receptor Associated Kinase 1; IS: immunosuppressive (i.e., corticosteroids alone or with other IS drugs); LOMG: late-onset (>50 years) MG; MAPK1: Mitogen-Activated Protein Kinase 1; MYB: MYB Proto-Oncogene, Transcription Factor; MuSK: muscle-specific kinase receptor; NFAT: Nuclear Factor Of Activated T Cells; oMG: ocular MG; PBMCs: peripheral blood mononuclear cells; RORγ: RAR-related orphan receptor gamma; RTX: rituximab; STAT3: Signal Transducer And Activator Of Transcription 3; TSLP: Thymic Stromal Lymphopoietin; WDR1: WD Repeat Domain 1. n.a.: not available.

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
