# Peer review of "Non-Coding RNAs in Myasthenia Gravis: From Immune Regulation to Personalized Medicine"

_cells, 2024, doi:10.3390/cells13181550_

Round 1
Reviewer 1 Report
Comments and Suggestions for Authors
Dear authors,
I have few recommendations of which I think they could improve the manuscript:
1. Please, add whether any literature / studies also provide data for children and adolescents. These data are important for the group of juvenile MG , too.
2. At the end in the conclusion and perspectives paragraph it would be helpful for the reader to get a more detailed list which miRNA should be investigated further in which group, e.g. as a potential biomarker. Otherwise it is too general and not that helpful for thinking next steps.
3. On page 8 3 figures close together are not comfortable to read this could be on 2 pages with text in between.
4. The definition of seronegative is not clearly described, there should be added a comment on other possibilities, e.g. cell based tests.
Author Response
We wish to thank Reviewer 1 for the thoughtful and useful comments that allowed us to improve our manuscript. Please find below our point-by-point reply.
- Literature data on the role of ncRNAs in pediatric and juvenile MG groups are very limited. To be more comprehensive in our description, we have now mentioned a very recent study that suggests a role of sEV-miRNAs as biomarkers and pathogenic factors in pediatric MG (Zhu et al. 2024; new ref. 102; page 11).
- We agree with the Reviewer’s comment on the utility to provide a list of miRNAs that should be investigated for their potential as disease biomarkers and targets of innovative therapies in MG. In the “Conclusion and perspectives” paragraph, we have now provided a set of MG-associated miRNAs as candidates to play such roles in MG (page 20).
- To avoid Table 1 and Figure 1 to be close together on page 8, we have now moved Figure 1 in a separate page and put text between them. As regards Figure 1, we think it is useful to maintain all together the three sections regarding thymus, PBMCs and serum/plasma, to provide immediate information on the ncRNAs DE in the three different compartments and allow a direct comparison among them (i.e. ncRNAs commonly altered in the different sites or specifically altered in thymus, PBMCs or serum/plasma).
- As suggested, we have now added in the Introduction and paragraph 2 some comments on seronegative MG and the ability of cell-based assays to detect autoantibodies undetectable by classical routine tests.
Reviewer 2 Report
Comments and Suggestions for Authors
This is a well written review on an interesting subject. The introduction is concise and on point. It was certainly a nice read. The tables and figures provide a very comprehensive overview. Well done!
Only minor comments:
- Maybe change the background of figure 1 to white. Its difficult to read as it is.
- 7. . Conclusions and future perspectives (Delete the point in the headline)
- Mabye also highlight that there are other current biomarker approaches to MG and highlight this avenue for the reader (without going into much detail). This might balance the manuscript a bit more.
Author Response
We wish to thank Reviewer 2 for the positive comments on our manuscript and the suggested revisions.
- In the revised article version, we have lightened the figure background for easier reading of the text.
- We have deleted the point in the paragraph 7 headline.
- As suggested, we have added a comment in the Conclusions (page 20) to highlight the role of genetic variants and circulating proteins as additional candidate disease biomarkers, along with ncRNAs, useful to promote personalised medicine in MG.
Round 2
Reviewer 1 Report
Comments and Suggestions for Authors
---